# Effect of In Situ Grown SiC Nanowires on the Pressureless Sintering of Heterophase Ceramics TaSi_2_-TaC-SiC

**DOI:** 10.3390/ma13153394

**Published:** 2020-07-31

**Authors:** Stepan Vorotilo, Evgeniy Patsera, Natalya Shvindina, Sergei Rupasov, Evgeniy Levashov

**Affiliations:** Department of Powder Metallurgy and Functional Coatings, National University of Science and Technology «MISiS», Russian Federation, Leninsky Prospect 4, Moscow 119049, Russia; patsera_yevgeniy@mail.ru (E.P.); natali19-03@list.ru (N.S.); vosapur@mail.ru (S.R.); levashov@shs.misis.ru (E.L.)

**Keywords:** tantalum disilicide, tantalum carbide, silicon carbide, nanowires, pressureless sintering, combustion synthesis

## Abstract

To ascertain the influence of SiC nanowires on sintering kinetics of heterophase ceramics, two composite powders (TaSi_2_-TaC-SiC and TaSi_2_-TaC-SiC-SiC_nanowire_) are fabricated by mechanically activated combustion synthesis of Ta-Si-C and Ta-Si-C-(C_2_F_4_) reactive mixtures. Remarkable compressibility is achieved for the TaSi_2_-TaC-SiC-SiC_nanowire_ composition (green density up to 84% as compared with 45.2% achieved for TaSi_2_-SiC-TaC) which is attributed to the lubricating effect of residual adsorbed fluorinated carbon (most likely C_4_F_8_). The outcomes of pressureless sintering of TaSi_2_-TaC-SiC and TaSi_2_-TaC-SiC-SiC_nanowire_ compositions are vastly different; the former experiences no significant densification or grain growth and does not attain structural integrity, whereas the latter achieves relative density up to 93% and hardness up to 11 GPa. The SiC nanowires are not retained in consolidated ceramics, but instead, act as a sintering aid and promote densification and grain growth. Sintering mechanisms of TaSi_2_-TaC-SiC and TaSi_2_-TaC-SiC-SiC_nanowire_ powders are analyzed using thermodynamic and ab initio grand potential calculations, as well as the analysis of grain size versus relative density relations. In the case of solid-state sintering, the densification and grain growth in heterophase non-oxide ceramics are governed by the same mechanisms as previously investigated single-phase oxides. The presence of SiC nanowires enhances grain-boundary related diffusion processes due to the high specific surface and aspect ratio of the nanowires. At 1500 °C, where the formation of the transient Si-based liquid phase is thermodynamically viable, only the SiC nanowire-containing composition demonstrated the intense grain coarsening and densification associated with liquid-assisted sintering. This effect can be attributed both to the presence of SiC nanowires and purification of residual oxide impurities due to C_2_F_4_-activated combustion synthesis employed for the in situ formation of SiC nanowires.

## 1. Introduction

Silicon carbide-based composites are widely used as structural ceramics due to their mechanical, thermal, and physical properties [1,2,3]. However, intrinsic brittleness of ceramics hinders the wider applications of SiC-based materials. This problem can be alleviated by reinforcing the ceramic matrix with SiC nanowires and whiskers, which can be produced by a variety of techniques [4,5,6,7,8,9] and introduced into the ceramic matrix using the powders metallurgy route. However, such an approach faces the following two major complications: (1) high cost or complicated methodology of SiC nanowire synthesis and (2) non-uniform distribution of SiC nanowires or whiskers in the powder mixtures and in the resulting sintered composites.

As an alternative, the concept of in situ growth of SiC nanowires within the ceramic matrix was suggested. The growth of SiC nanowires in SiC matrix was achieved by lanthanum-catalyzed re-deposition technology [10], laser processing [11], La_2_O_3_-Al_2_O_3_-catalyzed chemical vapor deposition [12,13], and chemically activated combustion synthesis [14].

The understanding of the effect of nanowires on the sintering kinetics and structural formation in ceramics is required to realize the potential of nanowire reinforcement [15]. Some insights can be gained from the studies of the sintering of single-phase ceramic nanopowders. A study of the sintering behavior of ZnO clusters reported a 118 K lower melting temperature as compared with bulk ZnO ceramics [16]. Numerous studies on TiO_2_ nanoparticles [17,18,19,20,21] have revealed that a decrease of particle size below 14 nm resulted in the stabilization of metastable anatase modification. Additionally, a nonlinear three-stage grain growth behavior has been reported, i.e., (1) slow growth of anatase nanoparticles up to phase transformation threshold (14 nm), (2) transformation of anatase into rutile, and (3) fast coarsening of rutile. The process of sintering was strongly affected by temperature and the initial orientation of nanoparticles.

The available data on the sintering mechanisms in heterophase ceramics are limited; however, the accelerating effect of nanowire-nanoparticle pairs on the neck formation has been demonstrated for heterogeneous oxide systems [22]. This could indicate the potential advantages of SiC nanowire as a reinforcing phase, and also as the sintering additive to promote grain growth, densification, and pore elimination in non-oxide ceramics [23,24].

The sintering of ceramics is generally governed by grain boundary diffusion, volume diffusion, surface diffusion, and evaporation-condensation mechanisms [25,26,27]. Grain coarsening results from surface diffusion and evaporation-condensation (e/c), whereas the densification is primarily governed by grain boundary diffusion and volume diffusion (trapped pore removal) [25,28,29,30,31,32,33]. The particle bonding and microstructural evolution lead to the shrinkage of sintered bodies and the formation of coherent solids [34,35,36,37,38,39]. The refinement of sintering components from micron- to nano-sizes can alter the contributions of the various sintering mechanisms by changing the area ratio between of vapor/solid surfaces and solid/solid interfaces [40,41,42,43,44]. 

The transition from single-phase to heterophase ceramics also introduces the possibility of the formation of interphase boundaries and dopant-stabilizing metastable phases. The mechanism of stabilization of metastable phases is not yet clear; however, the following trends were reported for the most extensively studied ZrO_2_-based models [45]: The decrease of grain size of metastable β-ZrO_2_ below 300 nm stabilized the phase and inhibited its spontaneous martensitic transformation and low-temperature degradation;The stability of β-ZrO_2_ showed a linear dependence on the concentration of stabilizing agents;The highest stability of β-ZrO_2_ was achieved by mixing trivalent and pentavalent stabilizers;Inhomogeneous distribution of stabilizer particles increased the susceptibility of β-ZrO_2_ to martensitic transformation;Slow cooling and isothermal dwelling during the heating-cooling cycles promoted the nucleation and growth of the thermodynamically stable α-ZrO_2_ phase;Dopant cations with radii smaller than Zr^4+^ reduced the lattice parameters, destabilized the fluorite structure, and promoted the formation of vacancies on non-metallic sublattice. As a result, the diffusion coefficients increased, lowering the ceramic solidification temperature and shortening the time of sintering, which allowed for the production of ceramics with a relative density close to one and enhanced mechanical properties.

The sintering of nanowire-reinforced ceramics was studied for single-phase oxide ceramic models. Hong-Fei Ju et al. [46] sintered ZnO rods and reported the sintering activation effect of structures with a high aspect ratio, which was previously discovered in silico for oxide nanowires [22]. The rods and the bulk of the ZnO ceramics demonstrated different behaviors in shrinkage, grain growth, and densification [46].

The topic of sintering of ceramic nano-objects (especially heterophase ones) is still mostly uncharted territory. Currently, there is very limited available data on the topics of sintering mechanisms of heterophase non-oxide nanoceramics, including SiC nanowire-reinforced compositions. The Ta-Si-C system is particularly interesting in this regard because it features a reversible phase reaction resulting in the formation of silicon-based melt [14]. Sintering techniques involving a transient liquid phase are widely used in ceramic manufacturing [45]; therefore, the investigation of the influence of SiC nanowires on the melt formation and overall sintering mechanisms in heterogeneous non-oxide compositions is both scientifically novel and practically applicable. The understanding and control of sintering behavior are pivotal for the engineering of ceramics with desirable properties [23,31,32,33,46].

In this work, two experimental models in the Ta-Si-C system were used for the investigation of the effect of SiC nanowires on the sintering behavior of heterogeneous ceramics, i.e., nanostructured SiC-TaSi_2_ ceramic composite and discretely-reinforced ceramic SiC-TaSi_2_-SiC_nanowire_. Experimental samples with uniform distribution of SiC nanowires were prepared using the recently developed (Ta + C_2_F_4_)-activated combustion route [14]. A temperature interval of 1300–1500 °C temperature was used for the study on the influence of in situ SiC nanowires on the kinetics and structural formation mechanisms in the case of solid sintering and sintering with the transient liquid phase.

## 2. Materials and Methods 

### 2.1. Calculations

#### 2.1.1. Calculations of Phase Stability (Ellingham Diagrams)

Three modules of the HSC Chemistry 6.0 software package (Metso Outotec, version 6.0, Espoo, Finland) were employed in this study. First, the “reaction equations” module was used for the calculation of changes in Gibbs free energy for various temperatures. Second, the “equilibrium compositions” module was employed to calculate the composition of each chemical at an equilibrium state. Third, the “Ellingham diagrams” module was used to summarize the phase stability of various phases as a function of temperature in the form of Ellingham diagrams. Solid lines represent the experimentally measured data and the dotted lines are projections of solid lines to higher temperatures.

#### 2.1.2. Construction of Grand Potential Phase Diagrams

To construct grand potential phase diagrams and to analyze the diffusion pathways in the Ta-Si-C system, the formation enthalpies of phases were calculated used mixed GGA and GGA + U (semiempirically-tuned generalized gradient approximations) frameworks [47]. Grand potential phase diagrams were constructed using PDApp software integrated into Materials API [48,49,50]. 

In grand potential phase diagrams, the effect of temperature and partial pressure/activity of a component is captured in a single chemical potential variable μ. More negative values correspond to higher temperatures and lower partial pressures of the element [51]. Diagrams provide useful insights on general trends and can predict the behavior of complex systems with reasonable accuracy [48,49,52].

The black nodes on a phase diagram represent phases that are stable under the given conditions. The black solid lines are projections of the convex hull construction into compositional space and form Gibbs triangles. An arbitrary composition at any point in the phase diagram other than the stable nodes will thermodynamically decompose to stable phases (black nodes) given by vertices of the triangle bounding that composition. The lever rule determines the relative proportion of each stable phase. Dotted lines on phase diagrams represent the projected diffusion pathways.

#### 2.1.3. Analysis of the Sintering Path via Grain Size Versus Relative Density Relationships

Two main phenomena that occur during the sintering of ceramics are densification, which arises from volume diffusion and grain boundary diffusion, and grain growth, which is related to the processes involving pores and grain boundaries. Relationships considering both densification and grain growth have been developed for ceramics based on the evolution of relative density and grain size [53].

For sintering densification, densification rate, *dD/dt,* and pore volume change rate, $\frac{d\sum_{p}^{T}}{dt}$, have the following relationship:(1)$$-\frac{dD}{dt}=\frac{1}{\sum_{g}}\frac{d\sum_{p}^{T}}{dt}$$
where *D* is the relative density of the sintered body, $\;\sum_{g}$ is the average grain volume, $\;\sum_{p}^{T}\;$ is the total pore volume in the sintered sample, and t is sintering time. The total pore volume change in the sintered sample can be determined by:(2)$$\frac{d\sum_{p}^{T}}{dt}=\Omega JN_{g}A$$
where *J* is the flux density of the material transferring to the pores from the center of a grain boundary, N_g_ is the average number of pores attached to a grain, *Ω* is the molar volume of the material, and A is the pore surface traversed by the material flux. A and J can be expressed as follows [46,53]:(3)$$J=-\frac{2D_{gb}\gamma_{sv}}{RTr_{p}\lambda }$$
(4)$$A=2\pi r_{p}\delta_{gb}$$
where *D_gb_* is the material diffusion coefficient along the grain boundary, *R* is the gas constant, γ*_sv_* is the solid/vapor surface energy, *r_p_* is the pore radius, *T* is the absolute temperature during sintering, δ*_gb_* is the grain boundary thickness, and λ is the grain boundary length.

The combinations of possible grain growth controlling mechanisms and densification controlling mechanisms allow one to establish the different grain size vs. relative density relationships, which are summarized in Table 1.

The integration constants are listed in Table 2. The calculations behind the equations and constants in Table 1 and Table 2 are described in detail elsewhere [53]. D* and *G** are, respectively, the starting relative density and starting grain size belonging to the sintering path, whereas *D* and *G* are the relative density and grain size measured in sintered specimens. The integration constants K1:K20 are related to material properties, grain boundaries, ambient pressure, sintering temperature, and mass diffusivity. Coefficients *k* in Table 2 have the dimension of a diffusion coefficient (m^2^ s^−1^); the subscripts and superscripts are related to the involved mechanisms (see abbreviations in Table 1).

To ascertain the governing sintering mechanism, grain size versus relative density graphs were plotted and fitted using the expressions in Table 1 with the constants from Table 2. The R^2^ value was used to assess the quality of fitting. The expressions with the highest R^2^ were considered as the best fit and signified that the associated mechanisms were responsible for densification and grain growth.

### 2.2. Experimental

#### 2.2.1. Preparation of Composite Powders

In this work, the following two heterophase experimental models in the Ta-Si-C system were prepared using mechanically activated combustion synthesis: TaSi_2_-TaC-SiC ceramic powders [54,55], designated TaSiC, and discretely-reinforced TaSi_2_-TaC-SiC-SiC_nanowire_ composition [14,56], designated TaSiC-nw.

The following powders were used for the preparation of reactive mixtures for combustion synthesis: TaPM grade tantalum powder (technical specifications TU 95.250-74, d < 74 µm; impurities (%) O 0.9; W 0.02; Ti, Mo, N, Al, and Cu 0.01 each; Co, Na, and Cr trace amounts); ball-milled 1A2 KDB grade semiconductor silicon (GOST 26239.1-84, d < 63 μm), P804T grade carbon black.

Previous investigations [14,56] have demonstrated that the highest quantity and quality of nanowires produced by the combustion of Ta-Si-C-C_2_F_4_ mixture was achieved if the combustion temperature of the mixtures was close to 1700 K. The overall temperature window (~200 K) was outlined, in which the growth of SiC nanowires was possible. The combustion temperature was controlled by adjusting the diameter of green pellets and the C:C_2_F_4_ ratio in the green mixture. The following optimal processing parameters were outlined for the reproducible synthesis of composites with uniformly-distributed SiC nanowires: preliminary mechanical activation of Ta-Si-C-C_2_F_4_ reactive mixtures (C:C_2_F_4_ ratio = 1:2) during 20 min in Aktivator-2S mill (JSC “Aktivator”, Novosibirsk, Russia), pressing of green pellets with a diameter 15 mm, relative density 60%, combustion in an SHS reactor in Ar atmosphere (P = 3 atm).

The following two reactive mixtures were prepared by high-energy ball milling (HEBM): (1) 53% Ta + 38% Si + 9% C and (2) 54% Ta + 33% Si + 3% C + 10% C_2_F_4_. The HEBM of reactive mixtures was conducted in an argon atmosphere (4 atm, 99.998%) using a double-station planetary ball mill Activator-2 (JSC “Aktivator”, Novosibirsk, Russia) equipped with steel mill and steel grinding medium (balls). Batches consisted of 20 g powders and were milled using 250 mL steel jars and 6 mm steel balls. The ball to powder mixture weight ratio was 20:1. The milling speed was 694 rpm at a rotational coefficient of K = 1. The total duration of the mechanical treatment was 20 min. The HEBM resulted in the formation of nanostructured composite particles with uniform distribution of reactants [54].

The reactive mixtures were pressed in a steel die to produce green pellets (d = 15 mm, relative density 60%). The individual pellets were placed in a laboratory combustion reactor SHS-20 (ISMAN, Chernogolovka, Russia), which was, then, vacuumed and, then, filled with gaseous argon up to 4 atm. The pellets were locally heated using a tungsten wire to initiate a chemical reaction with the subsequent propagation of a self-sustaining combustion front. Then, the combustion products were ball milled for 1 h at 60 rpm using WiseMixSBML mill (DAIHAN Scientific, Wonju-Si, South Korea) equipped with 250 mL steel jars and 6 mm steel balls (6:1 ball to powder mixture weight ratio). The duration of milling was optimized to achieve similar particle size distribution in both investigated ceramic compositions. The size of the resulting powders was determined via laser diffraction analysis using a Fritsch Analysette 22 Microtec Plus instrument (Fritsch, Idar-Oberstein, Germany). As-milled composite ceramic powers were subjected to magnetic separation using an LSV dry separator (SOLLAU, Velký Ořechov, Czech Republic) for the elimination of iron impurities. After the separation, the residual iron content in composite powders was below 1%. The phase composition of synthesized and sintered powders was measured by XRD analysis on Dron-4 installation (JSC “Burevestnik”, Saint-Petersburg, Russia) using CuKα radiation, 2θ range 10–110°, 0.1° step and 6 s exposition time. Spectra were treated using the JCPDS database.

#### 2.2.2. Pressing of Composite Ceramic Powders

Powders produced by milling of combustion products were subsequently pressed using a steel die (d = 12 mm) into pellets, which were, then, used for the sintering experiments. The relative density of the pellets (*D*_*_) produced using varied pressure was estimated using the geometric and weight measurements.

#### 2.2.3. Sintering Experiments

The different isothermal sintering runs constituting the experimental matrix were conducted in an electrical vacuum furnace Thermionic-T1 (JSC “Thermionic”, Podolsk, Russia). For each run, at least 10 samples were sintered concurrently. The sintering furnace cycle involved heating up at 50 °C/ min, holding at the dwelling temperature for a specific time, and free cooling to the room temperature.

The hydrostatic density of sintered specimens was measured on the AND1 GR-202 analytic weights (A&D, Tokyo, Japan) with a precision of 10^−4^ g. Helium pycnometer AccuPyc 1340 (Micromeritics, Norcross, GA, USA) was employed for true density measurement. Relative density was established as the ratio between the hydrostatic and the true densities.

#### 2.2.4. Microstructural Investigations and Mechanical Testing of Sintered Specimens

The structures of powdered and compact materials were studied by scanning electron microscopy (SEM) on an S-3400 N microscope (Hitachi, Tokyo, Japan) equipped with a NORAN energy-dispersive X-ray spectrometer (EDX) (Thermo Fisher Scientific, Waltham, MA, USA). The average grain size in the green samples, G_*, was measured from SEM micrographs of a milled powder, using a line-intercept method and taking into account at least 50 grains. A three-dimensional correction factor of 1.2 was used, meaning individual grains were approximated by spheres. The SiC nanowires were excluded from the statistic. The grain size in sintered specimens was measured similarly by SEM examination of the fractures and polished specimens. The hardness of sintered specimens was measured using an HSV-50 Vickers hardness tester (SHIMADZU Pte. Ltd., Columbia, MD, USA) using the load of 10 N.

## 3. Results and Discussion

Since the main focus of this study was on the analysis of the influence of combustion-induced SiC nanowires on sintering mechanisms of heterophase TaSi2-TaC-SiC ceramics, the phase composition and particle size distribution in compositions TaSiC and TaSiC-nw were controlled to ensure the highest achievable similarity between the two ceramic powders.

### 3.1. Characterization of the Ceramic Powders

The synthesized compositions, TaSiC and TaSiC-nw (Figure 1), were characterized by a similar phase composition (Figure 1b) and size distribution of composite powders after the combustion synthesis and ball milling (Figure 1c,d). Figure 1b provides the XRD spectrum for the TaSiC powders (phase composition 64 wt % SiC, 20 wt % TaSi_2_, 16 wt % TaC); the spectrum for TaSiC-nw powder (phase composition 58 wt % SiC, 22 wt % TaSi_2_, 20 wt % TaC) was similar, with slightly higher intensity of TaC peaks and lower intensity of TaSi_2_ peaks. The mean particle size of 8 and 6 µm was attained for TaSiC and TaSiC-nw compositions, respectively. Microstructure-wise, both TaSiC and TaSiC-nw powders were heterogeneous, i.e., every powder particle consisted of submicron-sized grains of TaSi_2_ and SiC (Figure 1e,f). In the case of TaSiC-nw composition, heterogeneous composite particles were surrounded by 5–10 µm long SiC nanowires. The mechanism of nanowire growth in this system was reported earlier [14,56].

Surprisingly, the TaSiC-nw powders demonstrated a nearly two-times higher compressibility (Figure 1a). Green bodies with a relative density as high as 78–84% were produced by conventional uniaxial single-sided pressing in steel dies at a pressure of 1–8 tons without the introduction of any lubricants. SiC nanowires are known for their high elasticity (up to 310 GPa) [57], and therefore one could expect that the SiC nanowire containing composition would demonstrate an increased elastic recovery, and therefore a decreased compressibility.

A previous investigation [14] revealed that the surface of SiC nanowires in the combustion products of Ta-Si-C-C_2_F_4_ reactive mixtures was covered by a thin (0.5–1 nm) carbon-based amorphous film. Earlier, Miyamoto [58] reported that Si-rich fluorinated SiC/CF_x_ films demonstrated a remarkable decrease in friction coefficient as compared with regular and fluorinated amorphous carbon films. Therefore, we attribute the increased compressibility of TaSiC-nw powders to the presence of adsorbed fluorinated carbon compounds. The nature of this compound and its possible influence on the subsequent sintering process are discussed in Section 3.4.

### 3.2. Sintering of TaSiC Composition: Thermodynamic Analysis

Concentration gradients and the resulting elemental fluxes have a major influence on the sintering of heterophase systems [59]. To ascertain the mechanisms responsible for the sintering of heterophase powders in the Ta-Si-C system, an isothermal section of the Ta-Si-C phase diagram of the system was constructed using the projections of the convex hull construction into compositional space (see Section 2.1.2). The ab initio calculated isothermal section (T = 0 K) of the Ta-Si-C phase diagram shows the existence of the following six stable phases: TaC, Ta_2_C, Ta_3_Si, Ta_5_Si_3_, TaSi_2_, and SiC, which is consistent with most of the published experimental data [60,61]. Some of the existing experimental reports suggested the presence of additional ternary phase Ta_5_Si_3_C_x_ in the system; however, the consensus was that this was not an independent phase but rather a solid solution of carbon in Ta_5_Si_3_ [60]. Therefore, this phase was not considered in further analysis. The synthesized compositions (TaSiC and TaSiC-nw) are both located in the TaSi_2_-TaC-SiC section of the diagram (Figure 2a). Therefore, three major chemical gradients are present in the ceramic composition and can result in the following: diffusion-driven mass transfer during the sintering (marked as dotted lines in Figure 2a), diffusion of Ta into silicon carbide (Figure 2b), diffusion of Si into tantalum carbide (Figure 2c), and diffusion of C into tantalum disilicide (Figure 2d). Diagrams in Figure 2b–d are constructed in relation to the chemical activity (µ) of the infusing element (Ta, Si, and C, respectively). Higher µ values correspond to lower local concentration of the element in question or lower temperatures.

If Ta acts as the principal diffusor in the system, the influx of Ta into TaSi_2_ + SiC and TaSi_2_ + TaC composite particles results in the decomposition of SiC with the formation of additional TaSi_2_ and TaC (Figure 2b). If the concentration of Ta increases or temperature decreases, TaSi_2_ is transformed, first, into Ta_5_Si_3_ and, then, into Ta_3_Si, whereas TaC forms a semicarbide Ta_2_C. The calculated Ta-potential phase diagram (Figure 2b) correlates very well with the existing studies on Ta/SiC diffusion pairs [62,63,64,65]. As was reported in the aforementioned studies, annealing of Ta/SiC pairs at 650–800 °C resulted in the formation of Ta_2_C (corresponds to µ_Ta_ > −11.8 eV, Figure 2b). The increase of annealing temperature to 900–950 °C prompted the formation of Ta_5_Si_3_ and Ta_2_C as the main reaction products (−12.499 < µ_Ta_ < −11.938 eV, Figure 2b). A further rise of annealing temperature (1000–1100 °C) produced TaC and Ta_3_Si_3_ (−12.656 < µ_Ta_ < −12.499 eV, Figure 2b), whereas at 1200 °C the reactive diffusion of Ta into SiC produced TaSi_2_ and TaC (−12.83 < µ_Ta_ < −12.656 eV, Figure 2b).

If Si acts as the principal diffusor in the system, the saturation of TaC by silicon results in the formation of TaSi_2_ and SiC (µ_Si_ > −5.655 eV, Figure 2c). The occurrence of this reaction at 800 °C has been reported in multiple experimental investigations on Si/TaC diffusion pairs [66,67,68,69,70,71,72]. In case C acts as the principal diffusor in the system, its diffusion into TaSi_2_ would result in the formation of the TaC and SiC phases (Figure 2d).

Since the grand potential phase diagrams are constructed at the temperature of 0 K, the stability of binary phases in the Ta-Si-C system was additionally evaluated using the Ellingham phase diagram (Figure 2e). SiO_2_ was added to the analysis since the EDS investigation revealed the presence of 2–3% oxygen impurities in TaSiC composition, and the previous investigation [55] has also revealed the formation of 2–5% amorphous SiO_2_ in the TaSi_2_-SiC combustion products. Studies on diffusion pairs in the Ta-Si-C system have also revealed the formation of stable amorphous SiO_2_ films [67,68]. The presence of such oxide impurities could significantly hamper the densification during the sintering since SiO_2_ remains the most thermodynamically stable phase in the system at temperatures up to 2000 °C (Figure 2e). The ΔG vs. T curves for SiC and tantalum silicide show clear inflection points at 1414 °C, which correspond to the melting point of Si (dotted line at Figure 2e). The occurrence of reversible phase transformations (Equations (15) and (16)) in the Ta-Si-C system has been previously suggested based on thermodynamic calculations [54] and corroborated experimentally by differential calorimetry and thermogravimetric studies, as well as by prolific formation of liquid phase during the hot pressing of TaSi_2_-SiC composites at temperatures above 1600 °C [55] as follows (Figure 2c):TaSi_2_ + SiC→ TaC+Si_melt_ (T > 1414 °C; µ_Si_ < −5.655 eV)(15)
TaC + Si_solid_ → TaSi_2_ + SiC (T < 1414 °C; µ_Si_ > −5.655 eV)(16)

The presence of transient Si-based liquid phase was associated with vastly improved sinterability of the TaSi_2_-SiC ceramics during the hot pressing [55], which is consistent with previously reported beneficial effects of transient liquid phase on the sintering of refractory ceramics. Improved densification is achieved due to the increased mass transport and anisotropic grain growth [73]. However, the sintering outcomes depend heavily on the kinetics of liquid-phase formation and wettability of solid phases by the as-formed melt [74]. The presence of oxide impurities often interferes with the reactive sintering. One reason is that most of the reduction processes are highly endothermic, and the Gibbs’s free energy becomes negative at temperatures that would promote densification (e.g., >1500 °C). Moreover, the reduction reactions can also lead to significant gas release, which interferes with densification. Additionally, the presence of oxides in the starting powders typically results in the partial retention of unreacted oxygen impurities in the final ceramic.

To investigate the interplay between the formation of Si-based transient liquid phase and oxide impurities and its effect on the sintering outcomes, three temperatures were selected for the vacuum sintering runs, i.e., 1300, 1400, and 1500 °C.

### 3.3. Sintering of TaSiC Composition: Microstructural Investigation and Mechanical Testing

Vacuum sintering of TaSiC powder at 1300–1500 °C, during 1–3 h, did not result in notable densification, grain growth, or formation of sintering-induced necks between the particles (Figure 3). Figure 3 provides the SEM images of fractured TaSiC specimens sintered at 1300 and 1500 °C for 3 h (relative density 45.5 and 50%, respectively). The green density of pressed green pellets pre-sintering was 45.2%; therefore, the maximum achieved densification was ~5%. Attempts of hardness testing were unsuccessful, even the indentation at 0.5 N loading destroyed the specimens. Multiple non-sintered agglomerates were present in the samples sintered at 1300 °C (Figure 3a,b) and 1500 °C (Figure 3c,d). Figure 3e provides histograms of the agglomerate size in TaSiC samples before sintering and after sintering at 1300 and 1500 °C measured by the random secant method. The histograms are nearly identical, which suggests the lack of agglomerate coarsening during the sintering. No evidence of transient liquid phase formation could be found. The statistics on grain size and relative density of sintered specimens are provided in Section 3.5.

The unsuccessful outcome of TaSiC sintering runs could be related to the inhibiting effect of oxide impurities on the Equation (15) since it requires clean phase boundaries between TaSi_2_ and SiC. Evidently, SiO_2_ acted as a diffusion barrier and inhibited the in situ formation of a transient liquid phase. This effect was considerably more pronounced in the case of pressureless sintering as compared with hot pressing; however, even in the case of hot pressing, higher sintering temperatures (1500–1600 °C) were necessary to achieve near pore-free ceramics in Ta-Si-C and Ta-Si-C-N systems [54,75], which tangentially corroborates the inhibiting effect of SiO_2_ impurities on the formation of a transient liquid phase in the system.

### 3.4. Sintering of TaSiC-nw Composition: Thermodynamic Analysis

Since the TaSiC-nw powders possessed remarkable compressibility which implies the presence of some form of fluorinated carbon, let us consider the phase equilibria of gaseous species during the combustion synthesis in the Ta-Si-C-O-F system (Figure 4a). The calculations of phase stability of gaseous species forming upon the decomposition of C_2_F_4_ in the presence of tantalum, silicon, and oxygen suggest that the most thermodynamically stable reaction products are TaF_5_, SiF_4_, and C_4_F_8_, which is consistent with previous experimental reports on the thermolysis of polytetrafluorethylene (C_2_F_4_) in vacuum and in the presence of oxygen [76,77,78,79]. The decomposition of C_2_F_4_ in the air was associated with the formation of CO_2_, CO, and COF_2_. The SiF_4_ and TaF_5_ phases are known to be the most stable fluorides of Si and Ta [80,81]. The melting and evaporation of TaF_5_ occurred at 95 and 229 °C, respectively. TaF_5_ can react with silicon (Equations (17) and (18)), tantalum disilicide (Euaqtion (19)), and silicon carbide (Equation (20)), but does not react with TaC. The reactions between TaF5 and Si, as well as TaF_5_ and TaSi_2,_ are thermodynamically favorable in the whole analyzed temperature interval (T = 25–2000 °C). Interestingly, the reaction of TaF_5_ with SiC does not occur at temperatures below 390 °C; however, at higher temperatures, the ΔG of the reaction decreases drastically (−991 kJ/mol at 400 °C, −1979 kJ/mol at 1000 °C) and surpasses Equations (17)–(19). Therefore, Equations (17)–(19) are likely to occur at temperatures below 400 °C, whereas at higher temperatures Equation (20) is predominant.
TaF_5_ + 1.25 Si → Ta + 1.25 SiF_4_ (−170< ΔG < −199 kJ/mol at 25 < T< 2 000 °C)(17)
TaF_5_ + 3.25 Si → TaSi_2_ + 1.25 SiF_4_ (−287 < ΔG < −255 kJ/mol at 25 < T < 2000 °C)(18)
TaF_5_ + 0.625 TaSi_2_ → 1.625 Ta + 1.25 SiF_4_ (−110 < ΔG < −165 kJ/mol at 25 < T < 2000 °C)(19)
4TaF_5_ + 5SiC = 5SiF_4_ + 4TaC + C (−255 < ΔG < −20419 kJ/mol at 390 < T < 2000 °C)(20)

SiF_4_ is gaseous at room temperatures and does not react with any other phase in the system at temperatures below 1000 °C; therefore, this phase evaporates as it forms (Equations (17)–(20)).

Among the stable fluorides, the C_4_F_8_ is likely responsible for the enhanced compressibility of TaSiC-nw powders (Figure 1a). C_4_H_8_ experiences thermal decomposition at 500–700 °C (Figure 4a). In the presence of SiO_2_, the decomposition of C_4_F_8_ likely leads to the formation of COF_2_, SiF_4_, and carbon oxides, which results in the elimination of oxide impurities (Figure 4b). Previous experimental studies have demonstrated that C_4_F_8_ is aggressive towards SiO_2_ at moderate and high temperatures [82], which has led to the utilization of C_4_F_8_ in plasma etching of silica (SiO_2_) [83].

### 3.5. Sintering of TaSiC-nw Composition: Microstructural Investigation and Mechanical Testing

Sintering of TaSiC-nw composition produced specimens with relative density up to 93%, which equates to 9% densification from the green state (84% relative density). The microstructural investigation of the specimen sintered at 1300 °C for 3 h (Figure 5a,b) revealed a more pronounced grain growth (up to 1.5 µm) as compared with the TaSiC composition sintered in the same conditions (Figure 3a,b). In the TaSiC-nw ceramics sintered at 1300 °C, the spheroidization of pores (0.5–1 µm) had already begun, although the networks of interconnected pores were still present (Figure 5b). An increase of the sintering temperature to 1400 °C resulted in a moderate increase in both relative density and grain size, whereas a further increase of the sintering temperature to 1500 °C produced pronounced changes in the microstructure (Figure 5c–e).

The size of the TaSi_2_ particles in specimens sintered at 1500 °C increased dramatically (up to 20 µm); the inspection of the recrystallized TaSi_2_ particles revealed that they were polycrystalline and contained precipitations of TaC phase (0.5–1 µm in size). The SiC grains remained an order of magnitude smaller than the TaSi_2_ particles (1–2 µm, Figure 5d,e). The pores in the microstructure (0.5–1 µm, Figure 5d,e) were rounded and isolated; no pore networks were found. The majority of pores were located on the grain boundaries (Figure 5d), whereas the number of pores within individual grains was very limited (Figure 5e). A previous study on the sintering features of ZnO rods found that the low ratio between average pore number per grain to average pore number per grain boundary was associated with the enhanced grain growth [46]. The pronounced coarsening of TaSi_2_ grains likely resulted from the presence of a transient liquid phase. Liquid phase sintering generally results in considerable grain growth, since the liquid diffusion grain growth rate constant is at least one order of magnitude higher than the solid diffusion grain growth rate constant [84]. Despite the drastic coarsening of individual TaSi_2_ particles, the overall microstructure of the sintered specimens remained uniform (Figure 5c). Hardness up to 11 GPa was achieved in TaSiC-nw specimens sintered at 1500 °C.

In the Ta-Si-C system, Si is expected to be the fastest diffusor at the Si/TaC interface up to 1414 °C [68]. The breaking of chemical bonds in the TaC compound is unlikely to occur, and therefore the solid-state diffusion of Ta in this system is improbable. The calculated activity diagrams (Figure 2b–d) support this assertion. The reaction most likely starts by Si in-diffusion into TaC via grain boundaries, followed by the dissociation of TaC, and the simultaneous formation of TaSi_2_ and SiC. Thus, carbon and tantalum would undergo only local rearrangement, which would not result in rapid grain growth. However, once the temperature of 1414 °C is reached and the Si-based liquid phase is formed (Equation (15)), an active rearrangement of Ta and C atoms becomes possible, leading to the fast coarsening of TaSi_2_ grains (Figure 5d,e). The amorphous oxide layers on the interfaces between the phases could act as a diffusion barrier and both slow down the diffusion of Si and inhibit the formation of Si melt (Equation (15)). In the case of TaSiC-nw composition sintered at 1500 °C, the oxide impurities are eliminated during the initial heating due to the release of adsorbed fluorocarbons (presumably C_4_F_8_, see Section 3.4). Oxide-free interfaces allow for the formation of Si melt at temperatures above 1414 °C. The presence of transient melt provides additional diffusion routes in the system (see Section 3.1) and results in the rapid densification of ceramic and growth and agglomeration of TaSi_2_ particles (Figure 5d,e).

To assess the densification mechanism, the relative density versus grain size curves were constructed for TaSi_2_ and SiC phases in TaSiC and TaSiC-nw compositions (Figure 6).

The statistics on TaC grains were not collected due to the low volume fraction of the TaC phase. In the case of the TaSiC composition, all the experimental points related to a given phase fall on one continued curve (brown curve for TaSi_2_, blue for SiC). Therefore, for TaSiC composition, any combination of sintering parameters (temperature and dwelling time) within the investigated range yields a similar sintered microstructure. For both TaSi_2_ and SiC, the grain size (G) is a linear function of the relative density (D), even including the (G*,D*) couple linked to the green state. Similar behavior has been previously reported during solid-state sintering of polycrystalline zirconia and yttria [46,53,85,86]. Therefore, one can conclude that the solid-state sintering, both in heterophase non-oxide ceramics and single-phase oxides, follows the same general trends as long as no phase interactions occur in the heterophase ceramics. Moreover, even if the phase interactions in heterophase ceramics are thermodynamically viable, the oxide layers can act as the diffusion barriers and impede the phase interactions.

One can see, however, that the size of the TaSi_2_ grains obtained in the specimen TaSiC-nw sintered at 1500 °C deviates from the curve obtained for 1400 and 1300 °C (Figure 6b). All the experimental points (Figure 6) were fitted with the different (densification mechanism and grain growth mechanism) relationships (Table 1 and Table 2). The resulting fitting coefficients (R^2^) are presented in Table 3.

The best fit for specimens TaSiC (sintering at 1300–1500 °C) and TaSiC (sintering at 1300–1400 °C) was obtained using Equation (5), meaning that densification is controlled by grain boundary diffusion and grain growth is controlled by the grain boundaries. Since the sintering path was identical, the difference in achieved relative densities can be attributed to the sintering-activating effect of SiC nanowires in the case of TaSiC-nw. For the TaSiC-nw composition sintered at 1500 °C, the best fits indicate that TaSi_2_ grain growth was governed by volume diffusion, whereas SiC grain growth resulted from diffusion along grain boundaries. This could explain the drastic difference in the size of TaSi_2_ and SiC particles in TaSiC-nw samples sintered at 1500 °C; however, such interpretation is tentative, since the employed relations were originally devised for solid-state sintering and might not fully reflect the processes occurring in transient liquid phases.

Therefore, in the case of solid-state sintering, the densification and grain growth in heterophase non-oxide ceramics were governed by the same mechanisms as previously investigated single-phase oxides [46,53]. The presence of SiC nanowires enhances grain-boundary related diffusion processes due to the high specific surface and aspect ratio of the nanowires. At 1500 °C, where the formation of the transient Si-based liquid phase is thermodynamically viable, only the SiC nanowire-containing composition demonstrated the intense grain coarsening and densification associated with liquid-assisted sintering. This effect can be attributed both to the presence of SiC nanowires and purification of residual oxide impurities due to C_2_F_4_-activated combustion synthesis employed for the in situ formation of SiC nanowires.

## 4. Conclusions

The following two heterophase ceramic compositions were prepared using mechanically activated combustion synthesis: TaSi_2_-TaC-SiC and TaSi_2_-TaC-SiC-SiC_nanowire_. The in situ formation of nanowires was achieved by the addition of a gasifying agent (C_2_F_4_) to the reactive mixture. The TaSi_2_-TaC-SiC-SiC_nanowire_ powders demonstrated improved compressibility (green density up to 84%), which was attributed to the presence of adsorbed fluorinated carbon (C_4_F_8_). During the pressureless sintering at 1300–1500 °C, the TaSi_2_-TaC-SiC specimen experienced a moderate increase in relative density (from 45.2 to 50%) and grain size (up to 0.5 µm) but did not attain structural integrity. In contrast, the sintering of TaSi_2_-TaC-SiC-SiC_nanowire_ composition produced relative density up to 93% and hardness up to 11 GPa, as well a pronounced grain growth of TaSi_2_ phase (up to 20 µm). The SiC nanowires were not retained in consolidated ceramics, but instead acted as a sintering aid and promoted densification and grain growth. Sintering mechanisms in TaSi_2_-TaC-SiC and TaSi_2_-TaC-SiC-SiC_nanowire_ were analyzed using ab initio grand potential calculations, thermodynamic investigations, and analysis of grain size versus relative density relations. The densification and grain growth during solid-state sintering of heterophase non-oxide ceramics are governed by the same mechanisms as previously investigated oxide models. The presence of SiC nanowires enhances grain-boundary related diffusion processes. At 1500 °C, only the SiC nanowire-containing composition demonstrated the intense grain coarsening and densification associated with liquid-assisted sintering as a result of either the presence of SiC nanowires or purification of residual oxide impurities by the fluorinated carbon released during the combustion of C_2_F_4_-containing reactive mixtures.

## Figures and Tables

**Figure 1 materials-13-03394-f001:**
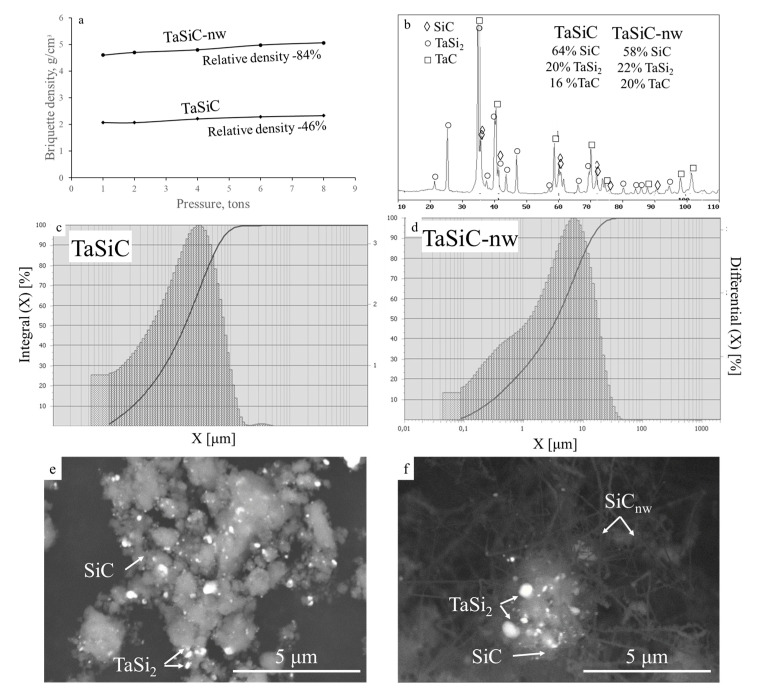
Compressibility curves (one-sided uniaxial pressing) of TaSi_2_-TaC-SiC ceramic powders (TaSiC) and discretely-reinforced TaSi_2_-TaC-SiC-SiC_nanowire_ composition (TaSiC-nw). (**a**) XRD spectrum for TaSiC powder and phase compositions (wt %) for TaSiC and TaSiC-nw powders; (**b**) Particles size distribution for TaSiC (**c**) and TaSiC-nw (**d**) powders; SEM images of TaSiC (**e**) and TaSiC-nw (**f**) powders after milling.

**Figure 2 materials-13-03394-f002:**
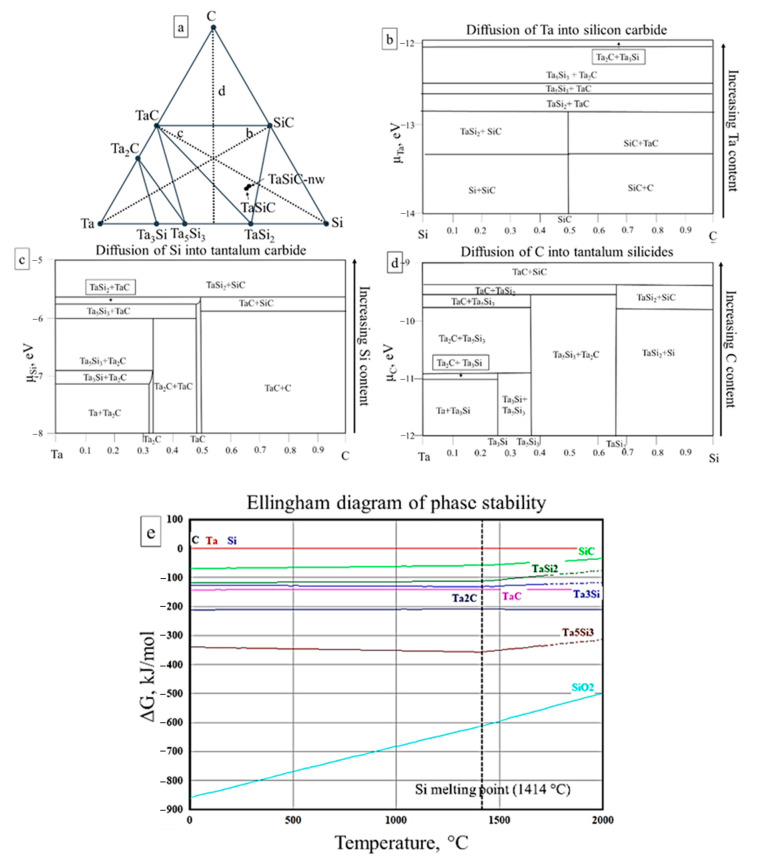
Ab initio calculated isothermal section (0 K) of Ta-Si-C phase diagram (**a**); Grand potential phase diagrams assuming Ta (**b**), Si (**c**), and C (**d**) as an open element/principal diffusor; (**e**) Ellingham diagram of phase stability in the Ta-Si-C system.

**Figure 3 materials-13-03394-f003:**
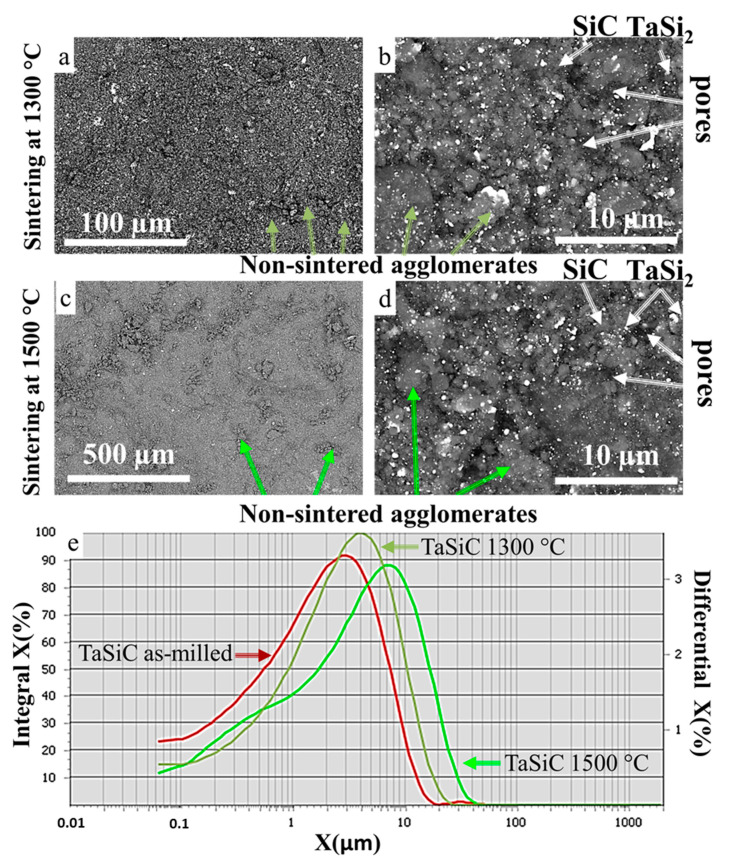
SEM images of TaSiC powders. (**a**,**b**) Sintered at 1300 °C; (**c**,**d**) Sintered at 1500 °C; (**e**) Histograms with the agglomerate size in the TaSiC samples before sintering and after sintering at 1300 and 1500 °C.

**Figure 4 materials-13-03394-f004:**
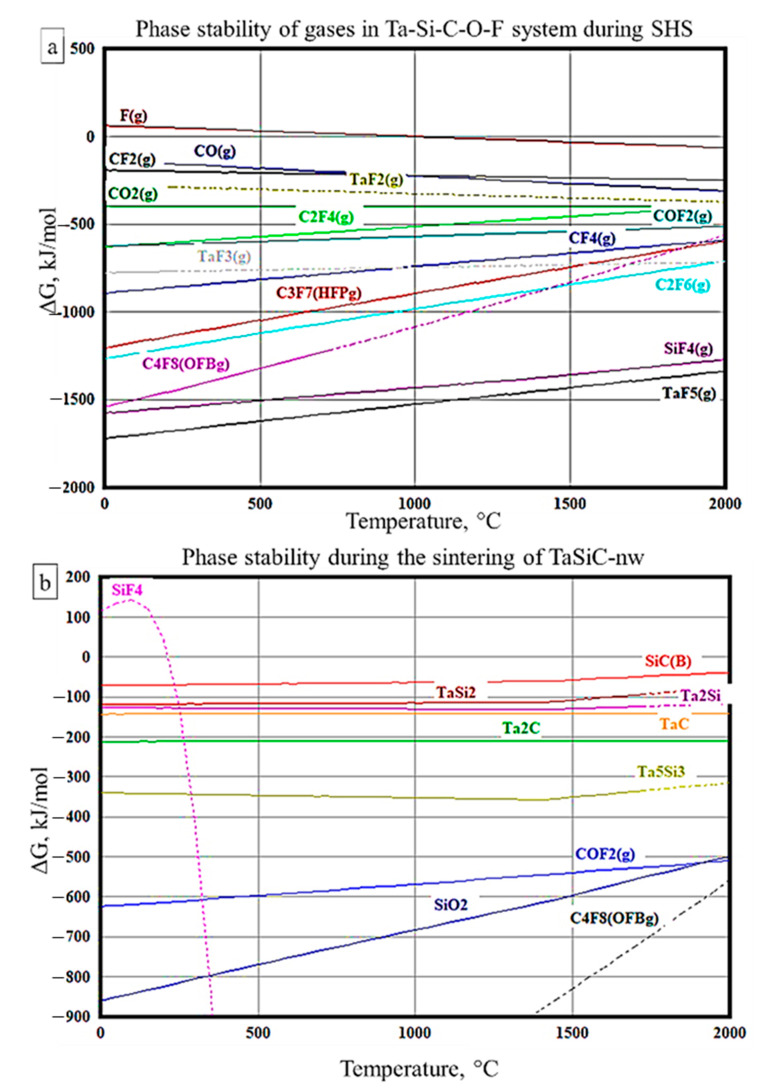
Ellingham diagrams of phase stability for selected gaseous species in the Ta-Si-C-O-F system (**a**) and sintering of TaSiC-nw composition (**b**). Solid lines correspond to experimentally measured values, dotted lines to projections.

**Figure 5 materials-13-03394-f005:**
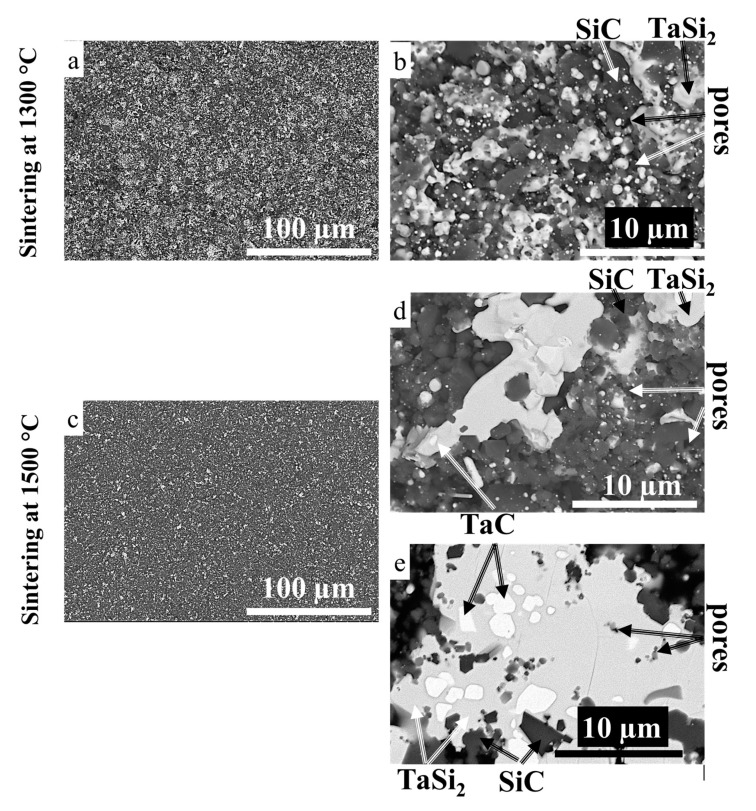
SEM images of TaSiC-nw composition. (**a**,**b**) Sintered at 1300 °C; (**c**,**d**,**e**) Sintered at 1500 °C, during 3 h.

**Figure 6 materials-13-03394-f006:**
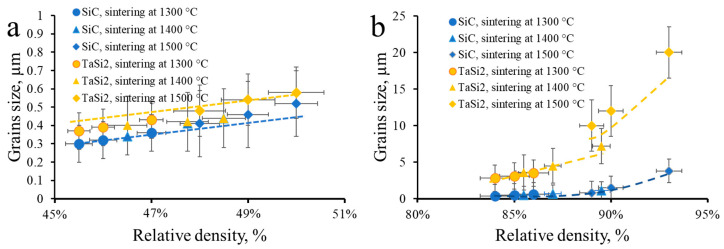
Grain size versus relative density curves. (**a**) For TaSiC composition; (**b**) For TaSiC-nw composition.

**Table 1 materials-13-03394-t001:** Grain size versus relative density relationships.

Grain Growth Mechanism	Densification Controlled by Grain Boundary (gb) Diffusion	Densification Controlled by Volume (v) Diffusion
Grain growth controlled by the grain boundaries (gb)	$\frac{1}{G^{2}}=-K_{1}D+\frac{1}{G_{*}^{2}}+K_{2}$ (5)	$\frac{1}{G}=K_{3}{\left(1-D\right)}^{2/3}+K_{4}$ (6)
Grain growth controlled by diffusion at the pores surface (s)	$G^{2}=K_{5}{\left(1-D\right)}^{-1/3}+K_{6}$ (7)	$G^{3}=K_{7}{\left(1-D\right)}^{2/3}+K_{8}$ (8)
Grain growth controlled by the pores with a bulk diffusion (d) pathway	$G=-K_{5}Ln\left(1-D\right)+K_{10}$ (9)	$G^{2}=K_{11}{\left(1-D\right)}^{-1/3}+K_{12}$ (10)
Grain growth controlled by the pores with a gas-phase (g) diffusion pathway plus an evaporation/condensation limiting step (e/c)	$LnG=-K_{13}{\left(1-D\right)}^{1/3}$ + $K_{14}$ (11)	$G=K_{15}ln\left(1-D\right)+K_{16}$ (12)
Grain growth controlled by the pores with a gas-phase (g) diffusion pathway plus a diffusion limiting step (d)	G=$-K_{17}Ln\left(1-D\right)+K_{18}$ (13)	$G^{2}=K_{19}{\left(1-D\right)}^{-1/3}+K_{20}$ (14)

**Table 2 materials-13-03394-t002:** Grain size versus relative density relationships, integration constants.

$K_{1}=\frac{2k_{gb}^{g}}{k_{gb}^{d}}$	$K_{6}=G_{*}^{2}-\frac{6k_{s}^{g}}{k_{gb}^{d}}{\left(1-D_{*}\right)}^{-1/3}$	$K_{11}=-\frac{6k_{v}^{g}}{k_{v}^{d}}$
$K_{2}=\frac{1}{G_{*}^{2}}+\frac{2k_{gb}^{g}}{k_{gb}^{d}}D_{*}$	$K_{7}=\frac{k_{s}^{g}}{2k_{v}^{d}}$	$K_{12}=G_{*}^{2}-\frac{6k_{v}^{g}}{k_{v}^{d}}{\left(1-D_{*}\right)}^{-1/3}$
$K_{3}=\frac{3k_{gb}^{g}}{k_{v}^{d}}$	$K_{8}=G_{*}^{3}-\frac{k_{s}^{g}}{2k_{v}^{d}}{\left(1-D_{*}\right)}^{-2/3}$	$K_{13}=-\frac{3k_{e/c}^{g}}{k_{gb}^{d}}$
$K_{4}=\frac{1}{G_{*}}-\frac{3k_{gb}^{g}}{2k_{gb}^{d}}{\left(1-D_{*}\right)}^{2/3}$	$K_{9}=-\frac{k_{v}^{g}}{k_{gb}^{d}}$	$K_{14}=lnG_{*}-\frac{3k_{e/c}^{g}}{k_{gb}^{d}}{\left(1-D_{*}\right)}^{1/3}$
$K_{5}=\frac{6k_{s}^{g}}{k_{gb}^{g}}$	$K_{10}=G_{*}-\frac{k_{v}^{g}}{k_{gb}^{d}}ln(1-D_{*})$	$K_{15}=-\frac{3k_{e/c}^{g}}{k_{v}^{d}}$
$K_{16}=G_{*}-\frac{k_{e/c}^{g}}{k_{v}^{d}}ln(1-D_{*})$	$K_{17}=-\frac{k_{g}^{g}}{k_{gb}^{d}}$	$K_{18}=G_{*}-\frac{k_{g}^{g}}{k_{gb}^{d}}ln(1-D_{*})$
$K_{19}=-\frac{6k_{g}^{g}}{k_{v}^{d}}$	$K_{20}=G_{*}^{2}-\frac{6k_{g}^{g}}{k_{v}^{d}}{\left(1-D_{*}\right)}^{-1/3}$	

**Table 3 materials-13-03394-t003:** Regression coefficient values (R^2^) for the different (densification mechanism and grain growth mechanism) couples.

Grain Growth Mechanism	Densification by Grain Boundary Diffusion		Densification by Volume Diffusion	
	TaSi_2_	SiC	TaSi_2_	SiC
**TaSiC composition, sintered at 1300–1500 °C**				
Grain growth controlled by grain boundaries	0.9926	0.9942	0.9813	0.9854
Grain growth controlled by pore surface diffusion	0.9341	0.9487	0.9494	0.9513
Grain growth controlled by pore volume diffusion pathway	0.9248	0.9273	0.9789	0.9183
Grain growth controlled by pore gas-phase diffusion with e/c control	0.9383	0.9339	0.9747	0.9284
Grain growth controlled by pore gas-phase diffusion pathway with diffusion control	0.9297	0.8679	0.9646	0.9175
**TaSiC-nw composition, sintered at 1300–1400 °C**				
Grain growth controlled by grain boundaries	0.9879	0.9903	0.9739	0.9630
Grain growth controlled by pore surface diffusion	0.9450	0.9504	0.9382	0.9429
Grain growth controlled by pore volume diffusion pathway	0.9203	0.9199	0.9378	0.9272
Grain growth controlled by pore gas-phase diffusion with evaporation/condensation control	0.9308	0.9272	0.9420	0.9507
Grain growth controlled by pore gas-phase diffusion pathway with diffusion control	0.9297	0.8679	0.9646	0.9175
**TaSiC-nw composition, sintered at 1500 °C**				
Grain growth controlled by grain boundaries	0.9236	0.9308	0.9739	0.9930
Grain growth controlled by pore surface diffusion	0.9551	0.9431	0.9382	0.9429
Grain growth controlled by pore volume diffusion pathway	0.9309	0.9123	0.9878	0.9572
Grain growth controlled by pore gas-phase diffusion with evaporation/condensation control	0.9528	0.9371	0.9423	0.9307
Grain growth controlled by pore gas-phase diffusion pathway with diffusion control	0.9197	0.8871	0.9541	0.9275

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
