# Peer review of "Effect of In Situ Grown SiC Nanowires on the Pressureless Sintering of Heterophase Ceramics TaSi2-TaC-SiC"

_materials, 2020, doi:10.3390/ma13153394_

Round 1

Reviewer 1 Report

In general the article is very good. The thermodynamic description is interesting, which brings a lot - especially when compared with the results from Ellingham charts. I recommend the article for acceptance after correcting 3 things:

(1) Please refer in the theoretical introduction to the formation of inter-phase boundaries and correlation with dopants stabilizing metastable phases; you write about zirconium oxide in your article, which is the most model example of this, it is worth describing the effect of additives on sintering, on real density and correlation with theoretical density, sintering mechanism, etc., all information can be found in this publication and its like:

Trends and perspectives in modification of zirconium oxide for a dental prosthetic applications - A review, Biocybernetics and Biomedical Engineering

(2) The appearance of the article is very spoiled by poor quality SEM photos with poorly marked markers and arrows, correct it because you are spoiling the article yourself, which is good and has a chance to cite,

(3) The same applies to Ellingham charts, a nice thing and you don't have much legibility and it may be useful to someone, improve the quality.

Author Response

Dear Reviewer,

We are thankful for your useful insights. Please find below the answers to the provided comments. 

Comment 1: In general the article is very good. The thermodynamic description is interesting, which brings a lot - especially when compared with the results from Ellingham charts. I recommend the article for acceptance after correcting 3 things:

Answer: We are deeply grateful for your high opinion about our manuscript and for your useful comments. We took care to address all the issues you’ve outlined. 

Comment: (1) Please refer in the theoretical introduction to the formation of inter-phase boundaries and correlation with dopants stabilizing metastable phases; you write about zirconium oxide in your article, which is the most model example of this, it is worth describing the effect of additives on sintering, on real density and correlation with theoretical density, sintering mechanism, etc., all information can be found in this publication and its like:

Trends and perspectives in modification of zirconium oxide for a dental prosthetic applications - A review, Biocybernetics and Biomedical Engineering

Answer: Thank you for the helpful suggestion. The issue of inter-phase boundaries and correlation with dopants stabilizing metastable phases was addressed in the introduction, including the references to the suggested article.

The transition from single-phase to heterophase ceramics also introduces the possibility of formation of inter-phase boundaries and dopant-stabilizing metastable phases. The mechanism of stabilization of metastable phases is not clear yet; however, the following trends were reported for the most extensively studied ZrO2-based models [45]:

  1. The decrease of grains size of metastable β-ZrO2 below 300 nm stabilized the phase and inhibited its spontaneous martensitic transformation and low-temperature degradation;
  2. The stability of β-ZrO2 showed a linear dependence on the concentration of stabilizing agents;
  3. The highest stability of β-ZrO2 was achieved by mixing trivalent and pentavalent stabilisers;
  4. Inhomogeneous distribution of stabiliser particles increased the susceptibility of β-ZrO2 to martensitic transformation;
  5. Slow cooling and isothermal dwelling during the heating-cooling cycles promoted the nucleation and growth of the thermodynamically stable α-ZrO2
  6. Dopant cations with radii smaller than Zr4+ reduced the lattice parameters, destabilised the fluorite structure and promoted the formation of vacancies on non-metallic sublattice. As a result, the diffusion coefficients increased, lowering the ceramic solidification temperature and shortening the time of sintering, which allowed for the production of ceramics with relative density close to 1 and enhanced mechanical properties.

Comment: (2) The appearance of the article is very spoiled by poor quality SEM photos with poorly marked markers and arrows, correct it because you are spoiling the article yourself, which is good and has a chance to cite,

Answer: Thanks for the helpful suggestion. We improved the quality of the pictures.

Comment: (3) The same applies to Ellingham charts, a nice thing and you don't have much legibility and it may be useful to someone, improve the quality.

Answer: The visual fidelity of the diagrams was improved, and their size was increased for better readability.  

Reviewer 2 Report

The paper contains the information of influence of SiC nanowires on sintering kinetics of heterophase ceramics. For measurements Authors fabricated two composite powders (TaSi2-TaC-SiC and TaSi2-TaC-SiC-SiCnanowire) by mechanically activated combustion synthesis of Ta-Si-C and Ta-Si-C-(C2F4) reactive mixtures. Sintering mechanisms of powder in systems TaSi2-TaC-SiC and TaSi2-TaC-SiC-SiCnanowire were analyzed using thermodynamic and ab initio grand potential calculations, as well as the analysis of grains size versus relative density relations. They observed increase in relative density, grains size, and mechanical properties in case of TaSi2-TaC-SiC-SiCnanowire specimen was attributed to facilitated densification during pressing and elimination of oxide impurities during sintering by adsorbed C4F8, enhancement of grain boundary diffusion by SiC nanowires and accelerated formation of transient liquid phase during the pressureless sintering.

In my opinion this paper can be interesting to readers of Materials journal. The paper is rather clearly presented. The paper contains a large number of reproduced figures and tables (6 figures and 3 tables). Most figures are legible and good quality. English of the paper is rather poor and do not meet the requirement of the journal – in my opinion the language of the paper should be improved. I am asking for corrections by a native speaker.

I also find some mistakes for example:

  • Affiliation: the scientific community knows what the abbreviation "MISiS" means in affiliation and where it is located, but it is necessary to specify the city and country.
  • Authors should correct Abstract – please describe the scientificity of the test results obtained.
  • Chapter Experimental – the names of the measuring equipment used should be given – model of equipment (manufacturer, city, country). Line: 176, 187, 190, 207, 208, 212 and 219.
  • In the whole paper, you write the values in percent as for example 78-84% (line 234) – you should write and value with unit with spaces (78-84 %).
  • In Figures 2 b, c, d and e – please increase the size of numbers and markings on axes.
  • In the whole paper, you write the values in percent as for example 1500°C (line 309) – you should write and value with unit with spaces (1500 °C).
  • In Figures 4 – no details are visible – please increase the size of chemical symbols and change color for more visible.
  • In Figures 6 – please change the range of ordinate (Y) axes – in the Figure 6a to 1.0 and in the case of Figure 6b to 25.
  • In my opinion Conclusions chapter should be changed (edited). In this chapter there are no summary of all significant research results obtained by the Authors and written in the Results chapter. The authors present the conclusions in the form of 3 paragraphs, without prior introduction to the performed tests.
  • Amount of references is also sufficient but some papers cited in the references (70 from all 85) are older then 10 years and 80 from all 85 are older then 5 years.
  • I the list of references I found 4 papers of the Authors of reviewed paper (first and corresponding Author). Are these the first studies related to this subject for the Authors?
  • The literature review (Introduction) is mainly based on very old publications. For this reason, the validity and scientific novelty of the research undertaken cannot be assessed.

The manuscript can be accepted for publication in Materials journal after MAJOR corrections.

Author Response

Dear Reviewer,

We are thankful for your useful insights. Please find below the answers to the provided comments. 

Comment: The paper contains the information of influence of SiC nanowires on sintering kinetics of heterophase ceramics. For measurements Authors fabricated two composite powders (TaSi2-TaC-SiC and TaSi2-TaC-SiC-SiCnanowire) by mechanically activated combustion synthesis of Ta-Si-C and Ta-Si-C-(C2F4) reactive mixtures. Sintering mechanisms of powder in systems TaSi2-TaC-SiC and TaSi2-TaC-SiC-SiCnanowire were analyzed using thermodynamic and ab initio grand potential calculations, as well as the analysis of grains size versus relative density relations. They observed increase in relative density, grains size, and mechanical properties in case of TaSi2-TaC-SiC-SiCnanowire specimen was attributed to facilitated densification during pressing and elimination of oxide impurities during sintering by adsorbed C4F8, enhancement of grain boundary diffusion by SiC nanowires and accelerated formation of transient liquid phase during the pressureless sintering.

In my opinion this paper can be interesting to readers of Materials journal. The paper is rather clearly presented. The paper contains a large number of reproduced figures and tables (6 figures and 3 tables). Most figures are legible and good quality. English of the paper is rather poor and do not meet the requirement of the journal – in my opinion the language of the paper should be improved. I am asking for corrections by a native speaker.

Answer: Thanks for the appreciation of the positive qualities of our manuscript. As suggested, the manuscript was corrected by a native speaker.

Comment: I also find some mistakes for example:

Affiliation: the scientific community knows what the abbreviation "MISiS" means in affiliation and where it is located, but it is necessary to specify the city and country.

Answer: The city and country were specified.

Comment: Authors should correct Abstract – please describe the scientificity of the test results obtained.

Answer: The abstract was corrected in line with the suggestions. The following passage was added:

In the case of solid state sintering the densification and grain growth in heterophase non-oxide ceramics are governed by the same mechanisms as previously investigated single-phase oxides. The presence of SiC nanowires enhances grain-boundary related diffusion processes due to the high specific surface and aspect ratio of the nanowires. At 1500 °C, where the formation of transient Si-based liquid phase is thermodynamically viable, only the SiC nanowire-containing composition demonstrated the intense grain coarsening and densification associated with liquid-assisted sintering. This effect can be attributed both to the presence of SiC nanowires and purification of residual oxide impurities due to C2F4-activated combustion synthesis employed for the in situ formation of SiC nanowires.

Comment: Chapter Experimental – the names of the measuring equipment used should be given – model of equipment (manufacturer, city, country). Line: 176, 187, 190, 207, 208, 212 and 219.

Answer: A more detailed information regarding the measurement equipment was provided.

Comment: In the whole paper, you write the values in percent as for example 78-84% (line 234) – you should write and value with unit with spaces (78-84 %).

Answer: The spaces were added between the values and percent units.

Comment: In Figures 2 b, c, d and e – please increase the size of numbers and markings on axes.

Answer: The requested corrections were made to the figures in question.

Comment: In the whole paper, you write the values in percent as for example 1500°C (line 309) – you should write and value with unit with spaces (1500 °C).

Answer: The spaces were added.

Comment: In Figures 4 – no details are visible – please increase the size of chemical symbols and change color for more visible.

Answer: The size of chemical symbols was increased, and color-coding was corrected for better visibility. Also, the size of Ellingham diagrams overall was increased.

Comment: In Figures 6 – please change the range of ordinate (Y) axes – in the Figure 6a to 1.0 and in the case of Figure 6b to 25.

Answer: The requested corrections were made to the figures in question.

Comment: In my opinion Conclusions chapter should be changed (edited). In this chapter there are no summary of all significant research results obtained by the Authors and written in the Results chapter. The authors present the conclusions in the form of 3 paragraphs, without prior introduction to the performed tests.

Answer: The Conclusion chapter was rewritten to provide a full coverage of the produced experimental and theoretical results.

Comment: Amount of references is also sufficient but some papers cited in the references (70 from all 85) are older then 10 years and 80 from all 85 are older then 5 years.

Answer: Unfortunately, the main body of works on the topic of this manuscript was published a while ago. Some newer articles were included in the references (for example, Nakonieczny, D. S., ZiÄ™bowicz, A., Paszenda, Z. K., & Krawczyk, C. (2017). Trends and perspectives in modification of zirconium oxide for a dental prosthetic applications – A review. Biocybernetics and Biomedical Engineering, 2017, 37(1), 229–245.).

Comment: I the list of references I found 4 papers of the Authors of reviewed paper (first and corresponding Author). Are these the first studies related to this subject for the Authors?

Answer: All of the cited papers were relevant to the current study: one addresses the combustion synthesis in Ta-Si-C system, another two provide a detailed description of the synthesis procedure for fabrication of in situ SiC nanowire-containing powders. The fourth paper provides a general overview of the combustion synthesis in the silicon-containing mixtures.  None of the cited papers contains the data on pressureless sintering of SiC nanowire-containing ceramics. 

Comment: The literature review (Introduction) is mainly based on very old publications. For this reason, the validity and scientific novelty of the research undertaken cannot be assessed.

Answer: Unfortunately, most of the works on the topic of this manuscript were published a while ago. Some newer articles were added; however, if the Reviewer has other papers in mind, we would gladly add them to the introduction of the paper.

Comment: The manuscript can be accepted for publication in Materials journal after MAJOR corrections.

Answer: We carefully addressed the issues outlined in the comments and hope that the quality of the manuscript improved as a result.

Reviewer 3 Report

"Improvement of pressureless sintering outcomes for heterophase ceramics TaSi2-TaC-SiC by in situ-grown SiC nanowires"

The Authors present and discuss results on the influence of SiC nanowires on the sintering kinetics of heterophase ceramics, namely for two composite powders - TaSi2-TaC-SiC and TaSi2-TaC-SiC-SiC(nanowire). 

The manuscript is well written, yet some aspects need careful revision; it should not be accepted for publication in this present form.

  • Title: not meaningful, rather hard to understand.
  • Abstract: meaningful; please use present tense (recommended).
  • Keywords: meaningful.

  1. Introduction
  • this section is comprehensive yet the Authors use long phrases, unnecessary examples, etc; the Authors must further insist on the importance and novelty of their work with respect to literature; further explain on your choice on this approach; furter present and discuss on the issues of heterophase ceramics.

  1. Materials and Methods
  • provide further and adequate references to the synthesis route, and further discuss on the parameters that need to be controlled.

  1. Results and discussion
  • the first paragraph needs to be some sort of introduction of the section: short, concise, explaining the main ideas of this work.
  • Figure 2. Ab initio calculated isothermal section (0 K) of Ta-Si-C phase diagram: please further discuss these results in text and compare to literature.
  • Figure 3. SEM: further discuss the results in text and consider to present a hystogram with the agglomerate size.
  • Figure 4. Ellingham diagrams: please further discuss these results in text and compare to literature.
  • Figure 5. SEM: further discuss the results in text; further explain the presence of pores with respect to literature, insist on their size / distribution / dispersion.
  • section 4. Discussion needs to be included here (3. Results and discussion).
  • Figure 6. Grains size versus relative density: further discuss these results in text and compare to literature.
  • the last paragraph of the section needs to be some sort of a conclusion (briefly presenting the main conclusion of this work)

  1. Conclusion
  • this section is meaningful, but please rephrase it to insist more on the novelty and importance of your work with respect to literature, and provide more on the actual and significant values / data.
  • remove the three points (recommended)

Minor aspects

The Authors need to avoid providing redundant data, superfluous text, or the repetitive use of speculative words (e.g. astonishing, outstanding, ...) or phrases.

To conclude, the manuscript should not be accepted for publication in its present form. Further revision is needed.

Author Response

Dear Reviewer,

We are thankful for your useful insights. Please find below the answers to the provided comments. 

Comment: The Authors present and discuss results on the influence of SiC nanowires on the sintering kinetics of heterophase ceramics, namely for two composite powders - TaSi2-TaC-SiC and TaSi2-TaC-SiC-SiC(nanowire). The manuscript is well written, yet some aspects need careful revision; it should not be accepted for publication in this present form.

Answer: We are thankful for your appreciation of our manuscript and took care to address all the outlined issues.  

Comment: Title: not meaningful, rather hard to understand.

Abstract: meaningful; please use present tense (recommended).

Keywords: meaningful.

Answer: The title was changed to “Effect of in situ-grown SiC nanowires on the pressureless sintering of heterophase ceramics TaSi2-TaC-SiC”. The abstract was rewritten in the present tense.

Comment: Introduction

this section is comprehensive yet the Authors use long phrases, unnecessary examples, etc; the Authors must further insist on the importance and novelty of their work with respect to literature; further explain on your choice on this approach; furter present and discuss on the issues of heterophase ceramics.

Answer: The Introduction section was partially abridged; additional discussion of the importance and novelty of our work was added. We also included a broader discussion of issues of heterophase ceramics, in particular regarding the formation of inter-phase boundaries and correlation with dopants-stabilizing metastable phases (which was also requested by another Reviewer).

The following passage was added:

The topic of sintering of ceramic nano-objects (especially heterophase ones) is still largely an uncharted territory. Currently there is very limited available data on the topics of sintering mechanisms of heterophase non-oxide nanoceramics, including SiC nanowire-reinforced compositions. Ta-Si-C system is particularly interesting in this regard, since it features a reversible phase reaction resulting in the formation of silicon-based melt [14]. Sintering techniques involving a transient liquid phase are widely used in ceramic manufacturing [45]; therefore, the investigation of the influence of SiC nanowires on the melt formation and on overall sintering mechanisms in heterogeneous non-oxide compositions is both scientifically novel and practically applicable.  The understanding and control of sintering behavior is pivotal for the engineering of ceramics with desirable properties  [23, 31-33,46].

Comment: Materials and Methods

provide further and adequate references to the synthesis route, and further discuss on the parameters that need to be controlled.

Answer: A more detailed description of the synthesis route was added to the Materials and Methods section.  

Previous investigations [14,56] demonstrated that the highest quantity and quality of nanowires produced by the combustion of Ta-Si-C-C2F4 mixture is achieved if the combustion temperature of the mixtures is close to 1700 K. The overall temperature window (~200 K) was outlined, in which the growth of SiC nanowires was possible. The combustion temperature was controlled by adjusting the diameter of green pellets and C:C2F4 ratio in the green mixture. The following optimal processing parameters were outlined for the reproducible synthesis of composites with uniformly-distributed SiC nanowires:  prelimilary mechanical activation of Ta-Si-C-C2F4 reactive mixtures (C:C2F4 ratio = 1:2) during 20 min in Aktivator-2S mill, pressing of green pellets with a diameter 15 mm, relative density 60%, combustion in an SHS reactor in Ar atmosphere (P=3 atm).       

Comment: Results and discussion

the first paragraph needs to be some sort of introduction of the section: short, concise, explaining the main ideas of this work.

Answer: The following paragraph was added to the beginning of the Results and Discussion section:

Since the main focus of the article is on the analysis of the influence of combustion-induced SiC nanowires on sintering mechanisms of heterophase ceramics, the phase composition and particles size distribution in compositions TaSiC and TaSiC-nw were controlled to ensure the highest achievable similarity between the two ceramic powders.

Comment: Figure 2. Ab initio calculated isothermal section (0 K) of Ta-Si-C phase diagram: please further discuss these results in text and compare to literature.

Answer: The following discussion was added:

Ab initio calculated isothermal section (T=0 K) of Ta-Si-C phase diagram shows the existence of six stable phases: TaC, Ta2C, Ta3Si, Ta5Si3, TaSi2, SiC, which is consistent with most of the published experimental data [60, 61]. Some of the existing experimental reports suggest the presence of additional ternary phase Ta5Si3Cx in the system; however, the general consensus is that this is not an independent phase but rather a solid solution of carbon in Ta5Si3 [60]. Therefore, this phase was not considered in further analysis

Figure 3. SEM: further discuss the results in text and consider to present a histogram with the agglomerate size.

Answer: The requested histograms were added (Figure 3e). The following discussion was added to the text: Multiple non-sintered agglomerates were present in the samples sintered at 1300 °C (Figure 3 a,b) and 1500 °C (Figure 3 c,d). Figure 3 (e) provides histograms of the agglomerate size in TaSiC samples before sintering and after sintering at 1300 and 1500 °C measured by the random secant method. The histograms are nearly identical, which suggests the lack of agglomerate coarsening during the sintering.

Figure 4. Ellingham diagrams: please further discuss these results in text and compare to literature.

Answer: The following discussion was added:

The calculations of phase stability of gaseous species forming upon the decomposition of C2F4 in presence of tantalum, silicon and oxygen suggest that the most thermodynamically stable reaction products are TaF5, SiF4 and C4F8, which is consistent with the previous experimental reports on the thermolysis of polytetraftoretylene (C2F4) in vacuum and in the presence of oxygen [76-79]. The decomposition of C2F4 in air was associated with the formation of CO2, CO, and COF2. SiF4 and TaF5 phases are known as the most stable fluorides of Si and Ta [80, 81]. The melting and evaporation of TaF5 occur at 95 and 229 °C, correspondingly.  TaF5 can react with silicon (Reactions 17,18), tantalum disilicide (Reaction 19) and silicon carbide (Reaction 20), but does not react with TaC. The reactions between TaF5 and Si, as well as TaF5 and TaSi2 are thermodynamically favorable in the whole analyzed temperature interval (T=25-2000 °C). Interestingly, the reaction of TaF5 with SiC does not occur at temperatures below 390°C; however, at higher temperatures the ΔG of the reaction decreases drastically (-991 kJ/mol at 400°C, -11284 kJ/mol at 500°C, -197932 kJ/mol at 1000°C) and surpasses the Reactions 17-19. Therefore, the Reactions (17-19) are likely to occur at temperatures below 400 °C, whereas at higher temperatures the Reaction 20 is predominant.       

TaF5 + 1.25 Si → Ta  + 1.25 SiF4 (-170<ΔG<-199 kJ/mol at 25<T<2000 °C)                                           (17)

TaF5 + 3.25 Si → TaSi2 + 1.25 SiF4 (-287<ΔG<-255 kJ/mol at 25<T<2000 °C)                                          (18)

TaF5 + 0.625 TaSi2 → 1.625 Ta + 1.25 SiF4 (-110<ΔG<-165 kJ/mol at 25<T<2000 °C)    (19)       

4TaF5 + 5SiC = 5SiF4 + 4TaC + C (-255<ΔG<-2041972 kJ/mol at 390<T<2000 °C)                       (20)

 SiF4 is gaseous at room temperatures and does not react with any other phase in the system at temperatures below 1000 °C; therefore, this phase evaporates as it forms (Reactions 17-20).  

Among the stable fluorides, the C4F8 is likely responsible for the enhanced compressibility of TaSiC-nw powders (Figure 1a). C4H8 experiences thermal decomposition at 500-700 °C (Figure 4a). In the presence of SiO2, the decomposition of C4F8 will likely lead to the formation of COF2, SiF4, and carbon oxides, which will result in the elimination of oxide impurities (Figure 4 b). Previous experimental studies demonstrated that C4F8 is aggressive towards SiO2 at moderate and high temperatures [82], which has led to utilization of C4F8 in plasma etching of silica (SiO2) [83].

Comment: Figure 5. SEM: further discuss the results in text; further explain the presence of pores with respect to literature, insist on their size / distribution / dispersion.

Answer: The following passage was added:

The majority of pores are located on the grain boundaries (Figure 5d), whereas the number of pores within individual grains is very limited (Figure 5e). A previous study on the sintering features of ZnO rods has found that the low ratio between average pore number per grain to average pore number per grain boundary is associated with the enhanced grain growth [46]. The pronounced coarsening of TaSi2 grains likely resulted from the presence of a transient liquid phase. Liquid phase sintering generally results in considerable grain growth, since the liquid diffusion grain growth rate constant is at least one order of magnitude higher than the solid diffusion grain growth rate constant [84]. Despite the drastic coarsening of individual TaSi2 particles, the overall microstructure of the sintered specimens remained uniform (Figure 5 c). Hardness up to 11 GPa was achieved in TaSiC-nw specimens sintered at 1500 °C.

In the Ta-Si-C system, Si is expected to be fastest diffusor at the Si/TaC interface up to 1414 °C [68]. The breaking of chemical bonds in the TaC compound is unlikely to occur, and therefore the solid-state diffusion of Ta in this system is improbable. The calculated activity diagrams (Figure 2 b-d) support this assertion. The reaction most likely starts by Si in-diffusion into TaC via grain boundaries, followed by the dissociation of TaC and simultaneous formation of TaSi2 and SiC. Thus, carbon and tantalum would undergo only local rearrangement, which cannot result in rapid grain growth. However, once the temperature of 1414 °C is reached and the Si-based liquid phase is formed (Reaction 15), an active rearrangement of Ta and C atoms becomes possible, leading to the fast coarsening of TaSi2 grains (Figure 5 d,e).

Comment: section 4. Discussion needs to be included here (3. Results and discussion).

Answer: The discussion was included in the section

Comment: Figure 6. Grains size versus relative density: further discuss these results in text and compare to literature.

Answer - the following discussion was added: Therefore, for TaSiC composition, any combination of sintering parameters (temperature, dwelling time) within the investigated range  will yield a similar sintered microstructure. For both TaSi2 and SiC the grains size (G) is a linear function of the relative density (D), even including the (G*,D*) couple linked to the green state. A similar behavior has been previously reported during solid-state sintering of polycrystalline zirconia and yttria [46,53,85,86]. Therefore, one can conclude that the solid-state sintering both in heterophase non-oxide ceramics and single-phase oxides follows the same general trends as long as no phase interactions occur in the heterophase ceramics Moreover, even if the phase interactions in heterophase ceramics are thermodynamically viable, the oxide layers can act as the diffusion barriers and impede the phase interactions.

Comment: the last paragraph of the section needs to be some sort of a conclusion (briefly presenting the main conclusion of this work)

Answer - the following paragraph was added:

Therefore, in the case of purely solid state sintering the densification and grain growth in heterophase non-oxide ceramics are governed by the same mechanisms as previously investigated single-phase oxides [46,53]. The presence of SiC nanowires enhances grain-boundary related diffusion processes due to the high specific surface and aspect ratio of the nanowires. At 1500 °C, where the formation of transient Si-based liquid phase is thermodynamically viable, only the SiC nanowire-containing composition demonstrated the intense grain coarsening and densification associated with liquid-assisted sintering. This effect can be attributed both to the presence of SiC nanowires and purification of residual oxide impurities due to C2F4-activated combustion synthesis employed for the in situ formation of SiC nanowires.

Comment: Conclusion

this section is meaningful, but please rephrase it to insist more on the novelty and importance of your work with respect to literature, and provide more on the actual and significant values / data.

remove the three points (recommended)

Answer: The conclusions were re-organized according to the suggestion.

Comment: Minor aspects

The Authors need to avoid providing redundant data, superfluous text, or the repetitive use of speculative words (e.g. astonishing, outstanding, ...) or phrases.

Answer: The superfluous text and speculative phraseology were excluded from the manuscript.

Comment: To conclude, the manuscript should not be accepted for publication in its present form. Further revision is needed.

Answer: we hope that we managed to correct the issues outlined in the review.

Round 2

Reviewer 2 Report

The authors have properly addressed the concerns from the referee. Most of my remarks have been included in the revised document.

The authors reformatted abstract, body text and conclusions of the paper. They corrected figures, descriptions, and captions of figures – their quality is satisfying. They corrected the names of the measuring equipment.

Referring to my substantive reservations – the authors made the necessary modifications. Authors have improved the language – language corrections are sufficient.

They corrected the list of references.

The manuscript can be accepted for publication in Materials journal in the current form.